# Neutrophil Extracellular Traps (NETs) in Ocular Diseases: An Update

**DOI:** 10.3390/biom12101440

**Published:** 2022-10-08

**Authors:** Jia Zeng, Min Wu, Yamei Zhou, Manhui Zhu, Xiaojuan Liu

**Affiliations:** 1Department of Pathogen Biology, Medical College, Nantong University, Nantong 226000, China; 2Department of Ophthalmology, Lixiang Eye Hospital of Soochow University, Suzhou 215000, China

**Keywords:** neutrophils extracellular traps, ocular diseases, inflammation

## Abstract

Neutrophil extracellular traps (NETs) are net-like complexes expelled from neutrophils, composing cell-free deoxyribonucleic acid (DNA), histones, and neutrophil granule proteins. Besides capturing and eliminating pathogens, NETs also exacerbate the inflammatory response associated with various diseases, including systemic lupus erythematosus, rheumatoid arthritis, and psoriasis. Currently, there are growing reports about NETs involved in the pathogenesis of ocular diseases. This review primarily focuses on the pathogenesis of NETs in the ophthalmology field, highlighting their importance in serving as potential targets for the therapy of ocular diseases.

## 1. Introduction

Neutrophils are abundant innate immune cells in the bloodstream undergoing functional changes when killing invading microorganisms in response to inflammatory stimuli. The most obvious ways of neutrophils response to pathogens include phagocytosis and degranulation [1]. In 2004, NETs was described as a novel bacterial killing mechanism of neutrophils, whereas the characteristics of NETs set them apart from apoptosis and necrosis [2]. Following priming with phorbol 12-myristate 13-acetate (PMA), which activates protein kinase C (PKC), NETs are generated by responsive neutrophils. During the formation of NETs, neutrophil chromatin decondenses, nuclear envelope disintegrates, cytoplasmic membrane breaks, extruded chromatin mixes with various antimicrobial proteins. Intense interactions occur between NETs and pathogens, including human immunodeficiency virus-1 (HIV-1) and Candida albicans [3,4,5]. Furthermore, accumulating studies describe that NETs act as a double-edged sword. On the one hand, NETs participate in a wide variety of functions of antimicrobial activities and resolution of inflammation [6]. Meanwhile, NETs play a pathologic role in multiple diseases, such as arthritis, systemic lupus erythematosus (SLE), and thrombosis [7,8,9].

## 2. Mechanism of NETs

### 2.1. Lytic NETosis

NETs are fragile structures via a new form of cell death that requires the disintegration of nuclear envelope and the decondensation of chromatin, subsequently, the cytoplasmic membrane breaks, decondensed chromatin, and granular contents release into the extracellular space, resulting in plasmatic membrane dissolution and neutrophil death. Meanwhile, NETs entrap bacteria fungi, protozoa, and viruses [10]. Defined components of suicidal NETs are neutrophil elastase (NE) and myeloperoxidase (MPO), which are stored in primary granules. NE is specifically designed to translocate from the cytoplasm to the nucleus, where it cleaves histones, promoting the unfolding of chromatin and the disruption of nuclear membrane [11]. MPO is also shown to be synergized with NE in this part, independent of its enzymatic activity [11]. Histones wrapping DNA around form nucleosomes and further organize into chromatin. While heterochromatin decondensation can be mediated by peptidylarginine deiminase 4 (PAD4), which plays an important role in innate immunity [12]. In addition, PAD4 is thought to prompt histone citrullination, contribute to the decondensation of chromatin and trigger the formation of NETs [13,14]. In other words, when activated by divalent calcium ion (Ca^2+^), PAD4 can reduce the strong positive charge of histones, which converts histone arginines to citrullines. After activation, NETs need reactive oxygen species (ROS) generated from nicotinamide adenine dinucleotide phosphate (NADPH) oxidase. NADPH oxidase leads to ROS generation, then in the reaction system superoxide anion can be catalyzed by superoxide dismutase (SOD) to produce hydrogen peroxide (H_2_O_2_) [10]. The H_2_O_2_ combines with MPO to generate hypochlorous acid (HOCl) and other oxidants, loosening histone-DNA interactions in a manner of chlorination of histones, which is similar to histone citrullination [15]. Furthermore, A. Hakkim et al. confirm that Raf-1 proto-oncogene serine/threonine kinase (c-Raf), mitogen-activated protein kinase kinase (MEK), protein kinase B (Akt), extracellular signal-regulated kinase (ERK), and PKC pathways are involved in the release of NETs (Figure 1). Additionally, a recent study further suggests that c-Raf-MEK-ERK pathway is upstream of NADPH oxidase [16]. The whole process takes approximately 2–4 h [6]. Besides PMA can induce strong and robust NETs responses, concanavalin A (ConA), living bacteria, ionomycin, fungi and some inflammatory cytokines such as interleukin (IL-6) and interleukin 8 (IL-8) induce NADPH oxidase-dependent NETs [17,18,19].

### 2.2. Vital NETosis

Earlier research conducted by F.H. Pilsczek et al. has reported a novel mechanism of NETs formation [20]. The process of vital NETs formation is rapid (5–60 min) which is induced by Staphylococcus aureus (*S. aureus*) through both complement receptors and Toll-like receptor 2 (TLR2) ligands, or by Escherichia coli (*E. coli*) directly via Toll-like receptor 4 (TLR4) or indirectly via TLR-4-activated platelets, as well as some specific bacterial products, such as lipopolysaccharide (LPS) via TLR4-activated platelets or complement proteins [6,21] (Figure 1). The decondensation of chromatin results in a double DNA helix appearing as “beads-on-a-string”, which is particular because, as has been observed, it appears only during NETs induced by *S. aureus*.

At the same time, *S. aureus* stimulates the production of NETs, which is much quicker than that of PMA [20]. In addition, it has been reported that neutrophils primed with *S. aureus*, then do not need the involvement of ROS produced by NADPH oxidase [20]. During this process, neutrophils release NETs without the breach of the plasma membrane or neutrophil lytic cellular death. Moreover, neutrophils can migrate and phagocytose [22,23]. Vital NETs are released by nuclear budding and vesicular transport correspondingly [24].

Another vital NETosis is triggered by mitochondrial DNA. S Yousefi et al. show that granulocyte-macrophage colony-stimulating factor (GM-CSF)-primed and component 5a (C5a)-stimulated neutrophils can release NETs, which is not associated with cell death but need the condition with the production of ROS [25].

The eye is composed of connective, vascular, and neural tissues. The connective tissue consists of clear and transparent cornea, serving as a primary barrier of host immunity to clear pathogens and keep healthy [26,27]. In the anterior segment, vascular tissue includes the choroid as well as two ciliary bodies, both of which are connected by the iris [27]. Traditionally, the neural tissue consisting of the retina involves the transport of electrical impulse to the brain [27].

Ocular surface releases inflammatory cytokines such as interleukin-1α (IL-1α), interleukin-1β (IL-1β), tumor necrosis factor-α (TNF-α), immunoglobulin A (IgA), IL-6, IL-8 and lactoferrin in tear fluids to regulate inflammation induced by microbes [28]. Neutrophils serve as the defensive front line of the ocular surface against an invading pathogen.

## 3. Cornea and Ocular Surface

It is well known that immunosuppressive molecules are recruited in the ocular microenvironment to influence the activity of immune cells. Inside the aqueous humor, there are numerous soluble immunomodulators, which are mixture composed of growth factors, cytokines, neuropeptides, and soluble receptors. The process of ocular immune privilege leads to apoptosis, boosts the production of anti-inflammatory cytokines, and mediates the activation of antigen-specific regulatory immunity [29].

The ocular surface is exposed continuously to the outer environment, which leads to the survival of different microbes on the ocular surface. There is a harmonious relationship between eye and commensal bacteria on the ocular surface [30]. Meanwhile, the commensal bacteria play an important role in both adaptive and innate immunity on the ocular surface [30]. A recent systematic review collects 11 published controlled cohorts, summarizing the most common bacteria or the core genera, showing that Corynebacterium appears in the all publications (11/11), then followed by Acinetobacter (9/11), Pseudomonas (8/11), Staphylococcus (7/11), Propionibacterium, and Streptococcus (both 5/11) [31].

### 3.1. Dry Eye Disease (DED)

Dry eye disease (DED) is recognized as a multifactorial dysfunction of the tears and ocular surface, which can lead to the symptoms of irritation, comprising visual disturbance, tear film instability, and increased tear film osmolarity. Moreover, it is accompanied by subacute inflammation of the ocular surface and neurosensory abnormalities, damage potentially accompany with ocular surface [32]. The prevalence of DED not only causes the subjects discomfort, but also it is more difficult for patients with DED to drive and read, as well as recreation. S. Koh et al. have demonstrated that under adapting to “new normal” pandemic coronavirus disease 2019 (COVID-19), as well as wearing a face mask more than 3 h a day, women with a history of prior DED, the symptoms will worsen [33].

Ocular surface inflammation is a peculiarity to definite DED. Episodic symptoms can be triggered by various factors, including environmental stresses, contact lens overwear, allergens, as well as the exacerbation of systemic autoimmune diseases [34]. In terms of the closed eye, the granulocyte-to-lymphocyte ratio (GLR) increases in patients with DED compared to that in normal individuals [35].

To date, inflammation has been recognized as a common etiopathogenic mechanism of DED comprehensively [36]. Cluster of differentiation 66b (CD66b) can be observed significantly elevated in patients with DED when they are closing their eyes. CD66b, expressed on the neutrophil plasma membrane as a marker associated with neutrophils secondary granule degranulation [37]. In some sense, among the risk factors for ocular surface inflammation, components such as extracellular DNA (eDNA) is considered as a crucial stressor that triggers the high level of pro-inflammatory cytokines, which leads to the amplified inflammatory damage [38]. For example, systemic immune-inflammation index (SII), which is involved in neutrophil, platelet, and lymphocyte counts, has increased in patients with a type of non-autoimmune DED [39]. Regarding the neutrophil-to-lymphocyte ratio (NLR), B. Sekeryapan et al. also find it increases in non-autoimmune patients with DED compared to that of normal subjects [40].

S. Tibrewal et al. describe that the DED severity degree is associated with higher tear osmolarity, and the increased NETs can be stimulated by hyperosmolarity [41]. In this context, tears of patients with DED display high levels of NETs, which has been demonstrated by various groups, including Sjogren’s syndrome and ocular Graft-versus-Host-Disease (oGVHD) [42,43,44,45]. For DED, NETs recruit when the eye is opening and accumulate in precorneal tear film, leading to ocular surface inflammation [43].

Evidence has suggested that DED can be induced by multiple factors. Meibomian gland dysfunction (MGD) belongs to one of the most common factors of DED [44,46]. MGD is often accompanied with altered lipid composition and tear film lipid layer disruption. NETs exert functions in both acute inflammation process and chronic tissue damage [46,47]. NETs are revealed to be accumulated in patients with blepharitis, which can be assessed by elevated C5a and IL-8 in tear fluid from patients with MGD, associated with extent of deficiency of tear fluid [48]. A Mahajan et al. report that NETs accumulating in the meibomian gland (MG) lead to duct occlusion and the acinar atrophy, based on the model of murine allergic eye disease (AED) [48]. Meanwhile, ocular surface diseases may result from molecular components of NETs. For example, histones can exert direct cytotoxicity on epithelial cells. Thus, the therapeutic use targeting the inhibition of PAD4 lead to restore the function of the MG [48]. In this context, other NETs-related makers such as the tear eDNA can be potential therapeutic targets for DED.

The dissonance of eDNA production and clearance [45] has been evaluated as a potential therapeutic target of DED. Deoxyribonucleases (DNases) can be used as a “waste-management enzyme” [49], acting as the extracellular endonuclease that degrades and clears DNA to form the backbone of NETs [45]. In the clinical trial, DNase eye drops possibly reduce inflammation-induced eDNA, corneal epitheliopathy, therefore, degrade corneal staining. Taking safety into consideration, DNase eye drops and placebo show no differences in adverse events, such as ocular infections [45].

Chronic graft-versus-host disease (GVHD) occurs after allogeneic stem cell transplantation, limiting the success of the procedure [50,51]. Ocular GVHD (oGVHD) is a severe ocular surface disease, due to dry eye is a major symptom of oGVHD. Inflammatory changes caused by oGVHD can affect multiple ocular tissues, including surface, cornea, conjunctiva, eyelids, and lacrimal glands. oGVHD shows not only clinical signs and symptoms of dry eye, but also dysregulated innate and adaptive immunity [42]. S. An and his team have profiled that in mucocellular aggregates (MCA) and ocular surface washings of patients with oGVHD, NETs are enriched [42]. The tears and ocular surface washings of patients with oGVHD show high levels of proteins related to NETs, such as NE, MPO, matrix metalloproteinase 9 (MMP-9), matrix metalloproteinase 8 (MMP-8), IL-8, TNFα, BDNF (brain-derived neurotrophic factor), oncostatin M (OSM), neutrophil gelatinase-associated lipocalin (NGAL) and tumor necrosis factor superfamily 14 (LIGHT/TNFSF14) [38,42]. Compared to none oGVHD and healthy subjects, patients with oGVHD present the symptom of corneal and conjunctival epitheliopathy and persistent corneal epithelial defects. In vitro and in vivo experiments suggest NETs-induced epithelial mesenchymal transition (EMT) causes epitheliopathy and delays epithelial wound healing. Meanwhile, NETs induce corneal and conjunctival fibrosis, immunoproliferation, and MGD.

S. An et al. demonstrate that a sub-anticoagulant dose of heparin (100 IU/mL) can dismantle NETs, thereby preventing epithelial, fibroblast, and alloreactive effects induced by NETs [42]. Furthermore, sub-anticoagulant dose heparin (100 IU/mL) displays non-toxic to the corneal epithelium [42]. Heparin translates protein-complexed DNA to protein-free DNA by displacing core histones from chromatin, providing a rationale for causing the destabilization of NETs [52,53]. DNase degrades DNA into smaller chains liberating bound NE. Subsequently, NE is inhibited by heparins to yield an anti-inflammatory effect [42]. The data indicate that using a combination of heparin with DNase I as a therapy strategy to clear the NETs from the ocular surface is possible. New biomarkers of NETs (eDNA, OSM, NGAL, and LIGHT/TNFSF14) show us the way to prevent NETs-induced epithelial, fibroblast, and alloreactive effects.

It is well known that a wide range of chemical and biological stimuli can induce the occurrence of NETs, however, short-chain fatty acids (SCFAs) as products mainly from bacterial fermentation of dietary fiber, which can be found in host tissues and blood [54]. Íñiguez-Gutiérrez and his workmates investigate the effect of SCFAs on the induction of NETs, demonstrating SCFAs induce the release of NETs in vitro, mediated by free fatty acid receptor 2 (FFA2R) [54]. Thus, SCFAs exert modulating effects on energy and endocrine response, and immune cell function [55,56]. The major components of SCFAs in the human body is acetate (C2), propionate (C3), and butyrate (C4), accounting for more than 95% [57]. Different intestinal microbes produce different SCFAs. For example, Bacteroidetes (Gram negative) mainly produce acetate and propionate, while butyrate is the final product are dominated by Firmicutes (Gram positive) [58].

It is known that the ocular microbiome contributes to DED. Li et al. investigated ocular bacterial diversity in humans with and without DED based on the 16S rRNA gene-based sequencing approach, among sequencing reads of subjects, the top three (Proteobacteria, Firmicutes, and Bacteroidetes) account for 92.47–100% [59]. Based on the results, we speculate that microbial communities on ocular surface may produce SCFAs, subsequently stimulate the release of NETs, and worsen the symptoms of DED.

Since the formation of NETs is mostly ROS dependent, in response to several biological and chemical stimuli. Vitamin C (VitC) is an essential vitamin for humans, as well as plays a role as an endogenous antioxidant. A previous study shows that VitC increases in millimolar quantities in polymorphonuclear neutrophils (PMNs), where it plays an important role in regulating neutrophil apoptosis [60,61]. Mohammed et al. recently showed that VitC attenuates neutrophilic capillaritis and decreases mortality in the murine sepsis model. Meanwhile, at a cellular level, VitC-deficient PMNs are more likely to activate endoplasmic reticulum (ER) stress and autophagy, which are vital for NETs [62]. It is remarkable that VitC also reduces NETs induced by PMNs and PMA from healthy human volunteers [63]. These results support the speculation that VitC can be useful for preventing and treating effects in diseases associated with ROS-dependent formation of NETs. Coincidentally, several antioxidative substances, such as catechin hydrate, epicatechin, rutin trihydrate, and the pharmacological substance 5-ASA, have the same effect on the inhibition of PMA-induced ROS production and NET formation [64].

### 3.2. Infectious Keratitis

#### 3.2.1. Bacteria Keratitis

Pseudomonas aeruginosa (*P. aeruginosa*) is a leading cause of bacterial keratitis with sight-threatening and rapidly progressive characteristics. Biofilm formed by *P. aeruginosa* leads to bacterial keratitis. Biofilm is formed by a type-3 secretion system (T3SS) that injects various virulence effectors (ExoS, ExoT, ExoY, and ExoU) into host cells, as well as bacterial Psl exopolysaccharide [65]. Biofilm formation primarily contributes to bacterial keratitis. Biofilm is deemed to perform frustrated penetrating neutrophils, due to the high expression of T3SS. In response, neutrophils release NETs, which form a barrier ‘‘dead zone” of DNA and degraded collagen, to confine bacteria to the external corneal environment and inhibit the dissemination of the bacteria to the brain [65].

Thymosin b4 (Tb4) is a natural and small peptide comprising 43 amino acids [66]. It is widely distributed and has been appreciated for multiple functions such as accelerates vascular endothelial cell proliferation, inhibits apoptosis, and ameliorates inflammation [66]. Investigators have used an experimental model of *P. aeruginosa*-induced keratitis, demonstrating adjunctive Tb4 treatment directly reduces the infiltration of PMNs, up-regulating the expression of anti-inflammatory markers, inhibiting ROS generation both in vivo and in vitro, down-regulating NETs, and regulating neutrophils apoptosis [67]. Overall, adjunctive Tb4 treatment influences neutrophils during the inflammatory response, reducing the infiltration of PMNs into the infected cornea and shifting the phenotype from pro-inflammatory to anti-inflammatory [67]. These studies will provide a novel visual for Tb4 in future clinical trials as a promising option for bacterial keratitis.

*S*. *aureus* is a common cause pathogen of ocular infections, which produces numerous toxins and enzymes capable of causing devastating vision loss. There is a report that said that S. *aureus* is isolated from 47.6% of blepharitis, 26.6% of conjunctivitis, and 25% of patients with keratitis [68]. Dysfunctional inflammation plays an important role during bacterial infections, which will cause host-induced inflammatory damage and vision loss during bacterial infection if it remains uncontrolled. Researchers find that during the early stage of bacterial infection in mice, neutrophils recruit to the site of infection from perilimbal circulation, foreshadowing the pathophysiology of acute-stage bacterial keratitis [69]. *S. aureus* 8325−4 is an α-toxin-positive parent strain, as α-toxin is deemed to be a virulence factor in some animal infection models and is significant for infections that disrupt epithelial barriers in the cornea. In the end, epithelial cell lysis led to underlying stroma exposure and increased neutrophil density [69]. β-toxin is a type of sphingomyelinase and is toxic to plenty of cells, such as fibroblasts, leukocytes, and macrophages. However, injecting intrastromally logarithmic-phase *S. aureus* into the cornea or using purified α- or β-toxins administered to normal eyes, then slit lamp examination and MPO activity of infiltrating PMNs are used to assess ocular pathology. The data suggest that α-toxin acts as a leading virulence factor during *S. aureus* keratitis, and β-toxin plays a role as a mediator of edema, showing a minor contributor to ocular damage [70].

#### 3.2.2. Fungal Keratitis

Fungal keratitis is a severe infectious disease that requires timely and appropriate treatment. The present study shows that neutrophils produce NETs in response to fungal keratitis [71]. The predominant causes of fungal keratitis are yeast (Candida) and filamentous fungi (Fusarium and Aspergillus) [72]. Additionally, long-term use of steroids has been reported to be a risk factor for fungal keratitis.

Glucocorticoids possess potent anti-inflammatory and immunosuppressive properties and have been widely used along with antibiotics for the treatment of a large variety of bacterial diseases [73]. However, using glucocorticoids in fungal keratitis may be controversial. Glucocorticoids are widely thought to play a vital role in regulating the host’s immunity and inflammation. F. Fan et al. established a mouse model of *Candida albicans* (*C. albicans*) keratitis, showing that compared with the control group, the dexamethasone (DXM)-treated group exacerbates the invasiveness of fungi by suppressing the formation of NETs [74]. It is intriguing to provide new insights into the understanding of the pathological mechanisms of fungal keratitis and the contradictory effect of glucocorticoids on fungal keratitis.

### 3.3. Cornea Injury

Ocular alkali burns are vision-threatening emergencies, while proper and effective clinical treatments promote corneal healing and decrease corneal opacity in the early period. A recent study shows that when NETs recruit for corneal capturing and killing pathogens, NETs play an important role in pro-inflammatory effects. The mammalian target of rapamycin (mTOR) is a ubiquitous serine/threonine kinase with the functions of maintaining homeostasis and acting as a master switch in the cell’s nutrient sensing pathways [75,76]. C.C. Yost et al. show that mTOR is of major relevance to inflammatory signals in neutrophils and is critical to mediate a novel pathway of post-transcriptional gene regulation. PMNs rapidly translate constitutive mRNA encoding retinoic acid receptor (RAR), which modulates expression of the chemokine IL-8 to mediate inflammatory events, while the translation of RAR is controlled by mTOR [77].

Recently, Yuan et al. showed that the localization of NETs can also be observed in the cornea, promoting inflammatory response and corneal neovascularization in the mouse corneal alkali burn model [78]. mTOR activity shows a negative effect on the formation of NETs. For example, A Itakura and his team suggest that pharmacological inhibition of mTOR signaling increases the formation of NETs via the expression of bacteria-derived peptide formyl-Met-Leu-Phe (fMLP) [75].

Acetylsalicylic acid (ASA) and DXM are used commonly and clinically as anti-inflammatory drugs, exerting their roles on the extracellular-acting form. Recently, more attention has been focused on the link between NETs and inflammatory injury after alkali burns. Wan et al. investigate the involvement of NETs in alkali burns, understanding the therapeutic effect of alkali-activated neutrophils (ANs) on corneal epithelial cell proliferation and migration, as well as assessing the regulatory effects of ASA and DXM on NETs [79]. C. Naffah de Souza and his team report the effect of pH on the formation of NETs. Alkaline pH promotes intracellular calcium influx, mitochondrial ROS (mROS) generation, PAD4-mediated CitH3, histone 4 cleavage, and finally increases NADPH oxidase (NOX)-independent formation of NETs [80]. Furthermore, NOX-dependent formation of NETs has been found [81]. A present study demonstrates that ASA, but not DXM, inhibits the alkali-induced dose- and time-dependent formation of NETs [79]. ASA treatment inhibits noncanonical nuclear factor kappa B (NF-κB) activation, not ROS, to decrease NETs formation, subsequently enhancing the migration of choroidal endothelial cells (CECs), which might be a promising strategy for improving the prognosis of corneal alkali burns [79].

## 4. Uveitis

Uveitis is known as a common, sight-threatening inflammatory ocular disease. Uveitis mainly affects the uvea, consisting of the iris, ciliary body, and choroid, as well as the retina, vitreous, and optic nerve [82]. Some factors are involved in the development of uveitis, such as genetics, age, sex, living habits, and geographical distribution [83]. Uveitis can be classified into two categories, infectious and non-infectious. Non-infectious uveitis is thought to be caused by an autoimmune or auto-inflammatory response. The effector CD4^+^ T cells, such as Th1 and Th17 cells, produce interleukin 17 (IL-17, also known as IL-17A) and interferon-γ (IFN-γ), which play a pathogenic role in the development of experimental autoimmune uveitis (EAU) [84,85]. Kallistatin (KBP, also named SERPINA4), which was originally identified from human serum, is identified as a member of the serine protease inhibitor (serpin) family. KBP exhibits various functions, including angiogenesis, oxidative stress, inflammation, apoptosis, fibrosis, and tumor growth [86]. A recent study indicates that KBP is directly responsible for the differentiation of Th17 cells and the production of IL-17, which is a key regulator of aggravating autoinflammation in EAU [87].

Pan and his workmates test the effect of temperature on the development of EAU. Compared to normal temperature (22 °C), the mice kept at a high temperature (30 °C) show worsened symptoms [88]. As MPO and NE are tested in serum and supernatants of neutrophils from EAU mice, the generation of NETs significantly elevates. Meanwhile, the numbers of Th1 and Th17 cells in association with IFN-γ and IL-17 mRNA up-regulate. This study illustrates that high environmental temperature increases the formation of NETs in EAU. That might be a promising target for the design of anti-inflammatory drugs. Significantly differential metabolites have been reported to play an independent cytokine-like role in immune regulation. For example, pro-inflammatory succinic acid worsens EAU induced by retinoid-binding protein (IRBP) in mice, stimulating the formation of NETs and the frequency of Th1/Th17 cells with increased production of IFN-γ/IL-17 [83,89,90].

Equine recurrent uveitis (ERU) is a common inflammatory ocular disease in horses. It is often described as a model for human autoimmune uveitis, owing to a number of similarities in clinical signs and pathogenesis [91,92]. There are investigations about vitreous body fluids (VBF), sera, and ocular histological sections from horses with or without ERU. Interestingly, several NETs markers in sera of ERU, such as free DNA, histone-complexes, and MPO, show a higher level in the samples from ERU horses compared to healthy horses, implying NETs are involved in the pathogenesis of ERU [93,94]. As the potential NETs-inducing factors, equine cathelicidins are up-regulated in the VBF of ERU-diseased horses. At the same time, cathelicidin-modulating factor IL-17 is a potential NET-inducing factor in ERU-diseased horses, while IL-17A and IL-17F are the closest genes to the single nucleotide polymorphism, which is linked with ERU [94,95]. These pro-inflammatory cytokines mediate the formation of NETs and the production of adenosine monophosphate (AMP), resulting in the occurrence of ERU. Leptospira, which induces NETs in a concentration-dependent manner in the VBF of ERU-diseased horses, is another contributor to the pathogenesis of ERU [93,96]. However, little is known about whether NETs contribute to the pathogenesis of ERU.

## 5. Diabetic Retinopathy (DR)

The International Diabetes Federation (IDF) declares that the global population with diabetes mellitus (DM) will be expected to increase from 463 million in 2019 to 700 million by 2045 [97]. About 80% of patients live in low- and middle-income countries (LMIC), such as China and India. DR is one of the leading causes of preventable partial or total vision impairment in adults [98,99].

DR has long been recognized as a microvascular disease. Chronic imbalanced sugar/glucose in the blood damages retinal microvascular [100]. Glycated end products, polyol accumulation, activation of PKC, and neutrophil-induced oxidative stress have been considered as contributing factors to DR [3,4]. Biochemical, hemorrhageal, and immunological mechanisms have been described to explain the vascular alterations in DR [101].

F. Binet and his team identify an inherent mechanism, providing new insight on NETs in retinopathy. Neutrophils, attracted by the senescent vascular units, release NETs, clear pathological retinal vasculature, and remodel retinal vasculature through apoptotic elimination of senescent endothelial cells (ECs) [102].

Inflammation, in particular, has been identified in recent studies, playing an essential role in the pathogenesis of DR. There is growing evidence that inflammation is independent pathophysiology of edema and pathogenic vascularization [103,104]. The causal relationship between inflammation to angiogenesis has been found during DR. Increasing vascular adhesion molecules, cytokines, chemokines, transcription factors, and growth factors-related inflammatory response are involved in DR development. These molecules stimulate activation and migration of leukocytes and leukostasis, subsequently capillary occlusion, retinal hypoxia, and endothelial cell damage [105].

Recently, it has been shown that patients with DM are exposed to increased oxidative stress (OS) induced by neutrophils, contributing to DR [106]. A recent report demonstrates NETs have been formed in patients with DR. Circulating NETs biomarkers can be used to estimate the disease risk of DR, such as circulating DNA-histone complex and polymorphonuclear NE [107]. DNA and histone, major components of NETs, activate factor XIIa. Then the factor XIIa activates the intrinsic coagulation pathway and the kallikrein-kinin system [108,109]. In vivo, the formation of NETs and the activation of factor XIIa are induced by glucose. D.Y. Song and workmates confirmed a hypothesis that DNA-histone complex and factor XIIa in patients with DR are elevated by hyperglycemia, which is related to the risk of DR and thrombotic complications [110]. It has been reported that ROS and vascular endothelial growth factor (VEGF, also named VEGF-A) interact mutually. ROS are critical chemical species in vascular endothelial growth factor receptor 2 (VEGFR2)-mediated signaling that leads to neovascularization in DR. Anti-VEGF therapy is valid for the reduction in NETs in this pathway [100]. The application of DNase I efficiently improves corneal nerve regeneration and corneal epithelial wound healing in diabetic mice [111].

## 6. Retinal Vein Occlusion (RVO)

Retinal vein occlusion (RVO) is the second most common retinal vascular disorder and results in irreversible visual damage [112]. Depending on the location of the occlusion, RVO can be divided into branch retinal vein occlusion (BRVO), hemiretinal vein occlusion (HRVO), and central retinal vein occlusion (CRVO). Although the mechanisms of RVO are still unclear, thrombosis inflammation, as well as oxidative stress, are believed to be the main causes of RVO. Local and systemic inflammation plays a role in retinal vein occlusion development by promoting atherosclerosis and inducing blood hypercoagulability [113]. The NLR has been used as a marker in a range of ophthalmic conditions such as DR and age-related macular [114,115]. Liu’s team makes a systematic review and meta-analysis, exploring the association of NLR and platelet-lymphocyte ratio (PLR) with RVO. Compared to controls, individuals with RVO have significantly elevated NLR and PLR [116]. Though the formation of NETs is considered to play pro-inflammatory and pro-thrombotic roles, the detailed relationship between NETs biomarkers and RVO progression remains indistinct. Wan and his team detect the concentrations of NETs biomarkers, including eDNA, MPO-DNA, and H3Cit, demonstrating that the circulating NETs-related markers significantly increase in patients with RVO [117]. These results demonstrate that NETs link inflammation to thrombus formation during RVO. The deposition of coagulant factors and fibrin can be stimulated by the procoagulant activity of NETs. Reversely, NETs formation is significant correlations with inflammation as well as thrombus. Crosstalk between NETs activation and thrombosis formation can be considered. The results provide NETs as potential prognostic and therapeutic targets for RVO in the future.

## 7. Age-Related Macular Degeneration (AMD)

Age-related macular degeneration (AMD) is the most common cause of blindness worldwide. Advanced AMD has two types, wet AMD and dry AMD [118]. Wet AMD is characterized by the presence of choroidal neovascularization (CNV), leading to retinal hemorrhage and exudation, and ultimately vision impairment. Dry AMD is characterized by advanced-stage geographic atrophy (GA), along with atrophy of the retinal pigment epithelium (RPE), choriocapillaris, and photoreceptors [119]. Metabolism, genetics, and environmental factors are involved in AMD. Local inflammation is associated with drusogenesis, RPE/photoreceptor degeneration, Bruch’s membrane (BM) disruption, and CNV progression. The secretion of pro-angiogenic factors can be enhanced by inflammatory cytokines [120]. Thus, inflammation contributes to the development of AMD.

Neutrophils and lymphocytes are important cellular components of the immune system. In their meta-analysis, S. Niazi and colleagues report that patients with AMD have a higher NLR, showing positive relevance between NLR and AMD [121]. In laser-induced CNV, neutrophils produce MMP-9 to degrade and remodel the extracellular matrix, a key process in angiogenesis [122]. Pro-angiogenic factors such as VEGF and IL-8 are secreted, which further recruit other immune cells and promote angiogenesis. Pro-inflammatory cytokine IL-8 triggers neutrophils to release NETs through the activation of Src and MAPK pathways [123]. Neutrophils release VEGF-A, hepatocyte growth factor (HGF), peptide 8 (Bv8), and MMP-9 to prompt angiogenesis. Furthermore, NETs directly promote angiogenesis [123]. A study has demonstrated that NETs exist in the eye with AMD, although there are no data about how NETs act on AMD.

## 8. Conclusions and Perspective

In summary, NETs play a crucial role in various ocular diseases (Figure 2). There is a ray of hope for therapeutic options by limiting the formation of NETs in ocular diseases (Table 1). However, considering that the pathogenesis of ocular diseases is intricate, the use of drugs such as eye drops to target NETs in some ocular diseases is likely to be safe, but there exist possible risks that need to be evaluated in a controlled clinical trial. In addition, distinct biological mechanisms are affected by different drug concentrations. It is worth noting that the mechanisms of NETs formation differ depending on the stimulus as well as the efficacy of treatment. Potential positive effects of NETs, including defense against pathogens and developing an appropriate balance of advantages and disadvantages, warrant further investigation. NETs are not the sole factor in the progression of ocular diseases, which contain complement cascade terminal products. Therefore, the effect of drug treatment on organism balance requires further investigation. There are no proper animal models for some ocular diseases; hence, follow-up investigations are difficult. NETs are significant as a target for ocular diseases. However, there is much work to be performed for using NETs-related biomarkers as diagnostic or therapeutic targets. The review primarily focuses on the activity of NETs, highlighting their ability to serve as a potential target for ocular disease.

## Figures and Tables

**Figure 1 biomolecules-12-01440-f001:**
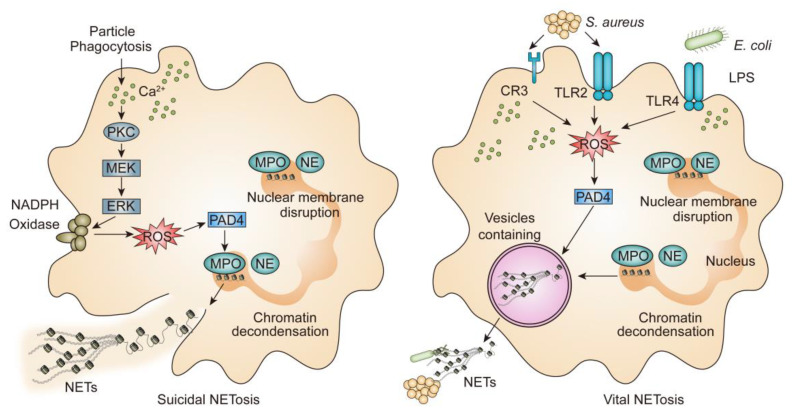
Mechanisms of NETs production were shown.

**Figure 2 biomolecules-12-01440-f002:**
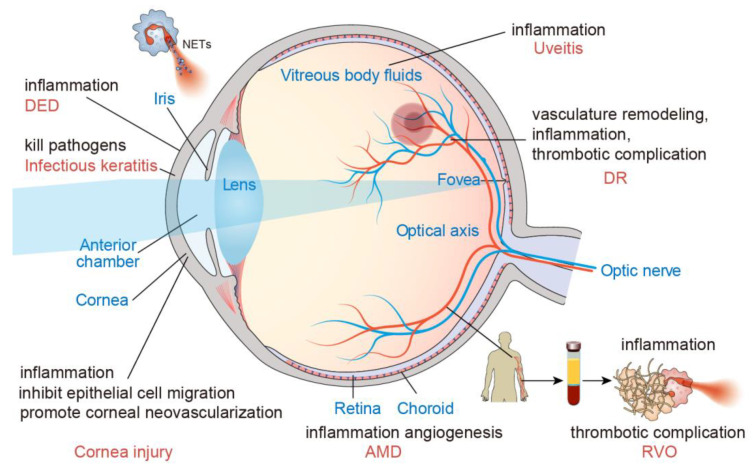
The localization and mechanisms of NETs in ocular diseases were shown.

**Table 1 biomolecules-12-01440-t001:** The interference of NETs in ocular diseases.

Ocular Disease	Cell and Animal Model or Patients	Treatment	Detection Methods	References
Dry eye (oGVHD)	Thy1-YFP mice	Heparin (100 IU/mL)	Heparin dismantled NETs, IF, HE staining, MTS assay, kinetic assay.	[42]
	Dry eye patients	0.1% DNase	OSDI; CGI; SGA; corneal staining, conjunctival staining; mucoid debris/strands; VBR grading scale.	[45,124]
	Type 1 diabetic mice model	DNase I	Corneal epithelial wound healing, IF, WB, ELISA, qRT-PCR; corneal reactive oxygen species; measurement of corneal mechanical sensitivity; corneal whole-mount staining.	[111]
*Pseudomonas* *aeruginosa* keratitis	*P. aeruginosa* keratitis mice model	DXM, tobramycin (Tobrex). Combination treatment	Ulcer area; density of opacity; corneal surface regularity; IF; scanning electron microscopy.	[125]
Ocular alkali burns	NaOH-stimulated neutrophils	Acetylsalicylic acid	Neutrophil-HCE adhesion assay; HCE proliferation and migration assay.	[79]
	Rabbit corneal alkali burn model	hAM-MSC	IOP determination; assessment of corneal thickness using ultrahigh-resolution OCT.	[126]
DR	Type 2 diabetic rat model	Anti-VEGF	ELISA; IF; qRT-PCR.	[100]

## Data Availability

The data used to support the findings of this study are available from the corresponding author upon request.

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
