# Peer review of "Neutrophil Extracellular Traps (NETs) in Ocular Diseases: An Update"

_biomolecules, 2022, doi:10.3390/biom12101440_

Round 1

Reviewer 1 Report

The review has almost all topics in ocular disease. The introduction is well done and  the treatments involved in the control of nets showed is interesting because is easy to read and is contain valuable  information to medical doctor and to basic researchers

1. The ocular enviroment is rich in immunosuppressive molecules to become an immune-privileged site, is necessary add a section about this topic or extend “3. Cornea and ocular surface”

2. In Section 3.2.1 Pseudomona is the main bacteria involved but another genus like  Staphilococcus must be included moreover the mechanism to avoid immune system is different ( only is mentioned in Netosis vital but what about in eye infection). Comment the eye microbioma and the mechanism to trigger NETs TLR-4 and TLR-2.

3. Other Microorganism molecules like short chain fatty acids are inductors of nets in vitro could be a potential source of nets in eye?

4. Oxidative stress trigger Nets mainly in suicide nets the use of topic antioxidants could be a new treatment? It has been proved VitC  in vitro for example

Author Response

Dear Editors and Reviewers:

Thank you for your letter and for the reviewers comments concerning our manuscript entitled “Neutrophil extracellular traps (NETs) in ocular diseases: an update” (No. biomolecules-1919542). These comments are all valuable and very helpful for revising and improving our paper. We have studied comments carefully and have made correction which we hope meet with approval. The main corrections in the paper and the responds to the reviewer's comments are as following. All changes have been highlighted.

Reviewer1:

  1. The ocular environment is rich in immunosuppressive molecules to become an immune-privileged site, is necessary add a section about this topic or extend “3. Cornea and ocular surface”.

The authors’ answer: In view of this valuable suggestion, we have added a paragraph about immune-privileged site of the ocular environment in the part“3. Cornea and ocular surface”. This part indicates the particularity of the eye. We think the supplementary paragraph does make the article more complete and help us to better discuss the following eye diseases.

  1. In Section 3.2.1 Pseudomona is the main bacteria involved but another genus like Staphilococcus must be included moreover the mechanism to avoid immune system is different ( only is mentioned in Netosis vital but what about in eye infection). Comment the eye microbioma and the mechanism to trigger NETs TLR-4 and TLR-2.

The authors’ answer: Thank you very much for your valuable suggestions after reviewing the full manuscript. Staphylococcus aureus and Pseudomonas aeruginosa are common species that invade the eye, and that discussion does not give a comprehensive picture of the review. Thus, we have added this section to the revised manuscript, as well as some common bacterial species in the eye and the mechanism to trigger TLR-4 and TLR-2 during NETosis.

  1. Other Microorganism molecules like short chain fatty acids are inductors of nets in vitro could be a potential source of nets in eye?

The authors’ answer: Your comment is really thoughtful, we admit that there are various bacteria in the ocular. Metabolites produced by various metabolisms may have an important effect on the eyes. Short chain fatty acids may be the source of NETs, which in turn have an impact on the occurrence and progression of the disease. We have fully discussed this point in the revised manuscript.

  1. Oxidative stress trigger NETs mainly in suicide nets the use of topic antioxidants could be a new treatment? It has been proved VitC in vitro for example

The authors’ answer: We accept your comments. As a common antioxidant, Vitamin C is very easy to obtain. It is amazing if it can be used in clinical treatment for NETs in some ocular diseases. We have added this part to the revised manuscript, hoping that the comprehensiveness of the review will be enhanced.

Reviewer 2 Report

The review of Jia et al., provides a comprehensive review that is focused on the pathogenesis of NETs in the context of eye disease.  There is a wealth of information for leukocyte biologists and clinicians. Although there are many reviews regarding NETs and neutrophil’s role in inflammatory diseases in general, this review provides a novel view towards ophthalmic diseases. A significant deficiency is the frequent grammatical errors throughout the manuscript. The review would also benefit from a figure that summarizes the complex mechanism of NET production and the various components as defined in the description of lytic NETosis (lines 47-87). In addition, earlier references to Figure 1 and Table 1 during the description of disease processes to highlight the association with NETs, where it occurs anatomically in the various disease processes would benefit the clarity.

Minor points.

-       Line 79 typo- “At the same time, S. aureus stimulates NETs to produce, the response is much quicker than that of PMA [20].”.

-       Awkward sentence- “Moreover, in this process, high integrity of neutrophils’ plasma membranes has been remained and still can migrate and phagocytose..”.

-       higher in DED subjects when they are closed eyes. Line 117

-       Incomplete sentence; “NETs exert functions in tissue damage and acute inflammation in the body..” [line 146].

Author Response

Dear Editors and Reviewers:

Thank you for your letter and for the reviewers comments concerning our manuscript entitled “Neutrophil extracellular traps (NETs) in ocular diseases: an update” (No. biomolecules-1919542). These comments are all valuable and very helpful for revising and improving our paper. We have studied comments carefully and have made correction which we hope meet with approval. The main corrections in the paper and the responds to the reviewer's comments are as following. All changes have been highlighted.

Reviewer 2

  1. A significant deficiency is the frequent grammatical errors throughout the manuscript.

The authors’ answer: Sorry for our negligence to make the grammatical mistakes. We have checked the manuscript and corrected the grammatical errors thoroughly, hoping this revised manuscript it makes it clear for you to read. 

  1. The review would also benefit from a figure that summarizes the complex mechanism of NET production and the various components as defined in the description of lytic NETosis (lines 47-87).

The authors’ answer: Thanks you for pointing out this problem for us. Additionally, we agree to you that adding a mechanism diagram will be easier for the readers to understand. According to your advice, we have drawn a mechanism figure about the complex mechanism of NET production and the various components as defined in the description of lytic NETosis in the end of the revised manuscript.

  1. Earlier references to Figure 1 and Table 1 during the description of disease processes to highlight the association with NETs, where it occurs anatomically in the various disease processes would benefit the clarity.

The authors’ answer: Thanks for your constructive opinion. The contents of Table 1 have been mentioned in the paragraphs of the review. The anatomical sites and the roles of NETs in different ocular diseases have been marked in Fig. 2.

  1. Line 79 typo- “At the same time, S. aureus stimulates NETs to produce, the response is much quicker than that of PMA [20].”.

The authors’ answer: Sorry for this mistake, and we have rearranged the sentences to make it easier to read.

  1. Awkward sentence- “Moreover, in this process, high integrity of neutrophils’ plasma membranes has been remained and still can migrate and phagocytose.”

The authors’ answer: We have rewritten the sentence to make it more logical.

  1. higher in DED subjects when they are closed eyes. Line 117

The authors’ answer: We are aware of the problem with this sentence, and we apologize for it. Therefore, we have rewritten this sentence for your better review.

  1. Incomplete sentence; “NETs exert functions in tissue damage and acute inflammation in the body..” [line 146].

The authors’ answer: We appreciate that you point out this mistake for us, due to incomplete sentence increases the difficulty in comprehension. Thus, we have corrected it.

Reviewer 3 Report

This manuscript is a comprehensive review, which focuses on how neutrophil extracellular trap contributes to the pathogenesis of ocular diseases. The topic of the review is clinically important and the relevant findings of the literature are discussed in a logical order. However, large amount of grammatical and phrasing errors make the understanding of the text difficult. Therefore, I suggest a thorough revision of the manuscript text with this regard.

Other specific suggestions:

1.       Due to the current preference in scientific literature of a non-stigmatizing language to describe diseases, adjectives should be avoided. Therefore, e.g. instead of "DED patients", "patients with DED" should be written. Please, amend this in lanes 106, 113, 117, 126, 129, 162, 163, 167, and 375.

2.       The sentences in lanes 56-58, 172-173, 177-179, 257-258, 382-383, 396-398, and 402-403 should be rephrased.

3.       The statements in lanes 152-154, 226-228, 296-298, 309-310, 341-342, 364-365, and 372-373 lacks reference(s).

4.       L87: Please, clarify what “ROS condition” means.

5.       L164: eDNA is not a protein.

6.       All abbreviations should be given when first mentioned in the text.

Author Response

Dear Editors and Reviewers:

Thank you for your letter and for the reviewers comments concerning our manuscript entitled “Neutrophil extracellular traps (NETs) in ocular diseases: an update” (No. biomolecules-1919542). These comments are all valuable and very helpful for revising and improving our paper. We have studied comments carefully and have made correction which we hope meet with approval. The main corrections in the paper and the responds to the reviewer's comments are as following. All changes have been highlighted.

  1. However, large amount of grammatical and phrasing errors make the understanding of the text difficult. Therefore, I suggest a thorough revision of the manuscript text with this regard.

The authors’ answer: We apologize for grammatical and phrasing errors in this manuscript. We have examined the manuscript and modified the grammatical and phrasing mistakes carefully. 

  1. Due to the current preference in scientific literature of a non-stigmatizing language to describe diseases, adjectives should be avoided. Therefore, e.g. instead of "DED patients", "patients with DED" should be written. Please, amend this in lanes 106, 113, 117, 126, 129, 162, 163, 167, and 375.

The authors’ answer: Thank you very much for your review. We accept your suggestions humbly. According to your requirements, we have found all the points that need to be changed in the full text to avoid the occurrence of such non-stigmatizing language.

  1. The sentences in lanes 56-58, 172-173, 177-179, 257-258, 382-383, 396-398, and 402-403 should be rephrased.

The authors’ answer: We express our appreciate to you for pointing out the existence of these problems for us carefully, and have rewritten the sentences you mentioned.

  1. The statements in lanes 152-154, 226-228, 296-298, 309-310, 341-342, 364-365, and 372-373 lacks reference(s).

The authors’ answer: Thank you for your patience in suggesting where we need to be corrected, we have inserted the references into the revised manuscript.

  1. L87: Please, clarify what “ROS condition” means.

The authors’ answer: Thank you for pointing out this error for us, and we have modified it in the revised manuscript.

  1. L164: eDNA is not a protein.

The authors’ answer: We have modified the expression error in the revised manuscript.

  1. All abbreviations should be given when first mentioned in the text.

The authors’ answer: We have checked the abbreviations in this manuscript, and supplemented the missing full name when the abbreviation appeared for the fir

Round 2

Reviewer 2 Report

Nicely edited, recommend acceptance.

Reviewer 3 Report

The manuscript was substantially improved. Therefore, I suggest the current version of this manuscript for publication.